# Balloon Eustachian Tuboplasty Combined or Not with Myringotomy in Eustachian Tube Dysfunction

**DOI:** 10.3390/jpm13111527

**Published:** 2023-10-25

**Authors:** Wei-Chieh Lin, Yao-Wen Chang, Ting-Ya Kang, Ciou-Nan Ye, Hung-Pin Wu, Chung-Ching Lin

**Affiliations:** 1Department of Otolaryngology, Head and Neck Surgery, Taichung Tzu Chi Hospital, Buddhist Tzu Chi Medical Foundation, Taichung 427, Taiwan; linweichieh1989@gmail.com (W.-C.L.); andyhorse11@gmail.com (Y.-W.C.); christinakang11611188@gmail.com (T.-Y.K.); joeyeh314@gmail.com (C.-N.Y.); hungpin_wu@yahoo.com.tw (H.-P.W.); 2School of Medicine, Tzu Chi University, Hualien 970, Taiwan

**Keywords:** Eustachian tube dysfunction, balloon dilation, myringotomy, ETDQ-7 scores

## Abstract

Background: Eustachian tube dysfunction (ETD) is a common disorder causing ear pressure, pain, and hearing loss. Balloon Eustachian tuboplasty (BET) is an emerging technique for dilating the Eustachian tube and treating ETD. Whether adding myringotomy improves BET efficacy is controversial. Methods: This retrospective study included 95 ETD patients undergoing BET alone (*n* = 44) or BET with myringotomy (BET + M; *n* = 51) between June 2020 and August 2021 at a single medical center. The primary outcome was the change in ETDQ-7 symptom scores from baseline to 6 months after treatment. Secondary outcomes included audiometry, endoscopy, Valsalva maneuver, and complications. Results: The ETDQ-7 scores improved significantly after treatment in both groups (*p* < 0.001), without significant between-group differences (*p* = 0.417). No significant differences occurred in the audiometry, endoscopy, and Valsalva results or in most complications between groups. One BET + M patient had a persistent tympanic membrane perforation. Conclusions: Both BET alone and BET + M effectively and safely improved the subjective and objective ETD outcomes. However, adding myringotomy did not further improve the outcomes over BET alone, while it incurred risks such as persistent perforation. BET alone may sufficiently treat ETD without requiring myringotomy in this cohort. Further randomized controlled trials should identify optimal candidates for BET alone versus combined approaches.

## 1. Introduction

Eustachian tube dysfunction (ETD) has become a prevalent ear, nose, and throat disorder that causes symptoms such as ear fullness, pain, and hearing loss, significantly impacting patients’ quality of life [1]. The pathogenesis of ETD may involve obstruction of the lumen, mucosal edema, impaired opening function, and other factors [2]. Treatment options for ETD include medications, Eustachian tube dilatation, myringotomy, and ventilation tube insertion. Among these, balloon dilation surgery is a minimally invasive procedure involving inflating a balloon within the Eustachian tube, and its use has increased in recent years [3]. Multiple studies have shown that balloon dilation surgery can significantly improve patients’ symptoms and has advantages such as minimal surgical trauma [4,5]. However, there is ongoing debate over whether BET should be combined with myringotomy, with some ETD patients undergoing BET plus myringotomy and others undergoing BET alone [6,7]. Currently, there is no definitive consensus on whether adding myringotomy provides additional benefit over BET alone for ETD treatment. Further research is needed to determine the optimal use of myringotomy with BET versus BET alone in ETD patient populations.

Myringotomy is a traditional surgical treatment for ETD that can immediately release pressure and fluid buildup in the middle ear to relieve symptoms [8]. However, the benefits of myringotomy may only be temporary before the fluid reaccumulates [9,10]. Myringotomy also carries risks such as infection and persistent tympanic membrane perforation [9,11,12,13]. In comparison, techniques such as balloon dilation may more directly treat the pathological cause of ETD [14]. Therefore, there is considerable debate over whether myringotomy improves the efficacy of balloon dilation [15,16,17].

The aim of this study was to compare the therapeutic effects and safety outcomes of balloon dilation with or without myringotomy for Eustachian tube dysfunction. The results can help provide evidence-based recommendations for the optimal surgical approach in managing ETD.

## 2. Materials and Methods

### 2.1. Ethical Consideration

This retrospective clinical trial was granted expedited ethical approval under protocol REC110-16 by the institutional research ethics board. Throughout the study, rigorous adherence to the ethical principles outlined in the Declaration of Helsinki ensured that the rights and welfare of all participants were protected.

### 2.2. Experimental Design

The study population comprised patients who underwent balloon Eustachian tuboplasty (BET), with or without concurrent myringotomy, at a single medical center between June 2020 and August 2021. Patients were included if they had Eustachian tube dysfunction (ETD) symptoms continuously for over 3 months and met the following criteria: scored above 14 points on the validated Chinese version of the 7-item Eustachian Tube Dysfunction Questionnaire (ETDQ-7), reflecting subjectively severe ETD symptoms [15,16]; had objectively poor or moderate Eustachian tube function on standardized assessments [17].

The exclusion criteria were as follows: history of ear surgery or malignancy; acute or chronic otitis media, otitis media with effusion, or congenital ear or nasal anomaly; recent use of ototoxic medication; pregnancy.

### 2.3. Preoperative Protocol

All patients were evaluated preoperatively using a tympanogram, ETDQ-7 score, and Eustachian tube function test. There was education on the Valsalva maneuver for daily practice. Nasopharyngoscopy was performed to evaluate the Eustachian tube mucosa inflammation scale [18]. Otoscopy was performed to exclude eardrum perforation or otitis media with effusion [17].

### 2.4. Surgical Techniques

All BET procedures were performed under general anesthesia. During the surgery, the physician first inserted a balloon catheter through the patient’s nasal cavity into the cartilaginous portion of the Eustachian tube under direct endoscopic visualization. Abnormal Eustachian tube anatomy can increase surgical difficulty and introduce complication risks. Therefore, before catheter insertion, the physician carefully inspected the patient’s Eustachian tube to ensure a smooth and safe placement. Once the balloon catheter was positioned appropriately, the physician used a sterile syringe to inflate the balloon with purified sterile distilled water to a pressure of 10 bar (1000 kPa). This first inflation cycle lasted 2 min, allowing the balloon to fully dilate the Eustachian tube lumen and improve its structural integrity. The balloon then rested for 30 s before a second inflation cycle to 10 bar pressure for another 2 min [3]. The interval rest period between inflations allowed the Eustachian tube mucosal tissues to gradually accommodate the dilation, reducing mucosal trauma. Throughout the balloon inflation process, the physician closely monitored the balloon pressure and Eustachian tube status to prevent complications. Precise technique and vigilant pressure control are crucial to minimize risks like barotrauma, mucosal injury, or improper balloon placement during BET procedures.

For patients also undergoing myringotomy, steps were taken before completing the BET. The patient’s head was rotated to the opposite side to better expose the surgical ear. Before making the incision, the physician first cleaned the external auditory canal using gentle suction to remove any earwax, debris, or secretions obstructing visualization. Anomalous ear canal anatomy can also increase the surgical difficulty; thus, the canal was thoroughly inspected. With a clear view of the tympanic membrane under the surgical microscope, a small precise radial incision was made in the thin cartilaginous region of the anteroinferior quadrant.

### 2.5. Postoperative Care Protocol

All patients were discharged the day after surgery. They were prescribed a 7-day course of oral systemic antihistamines to reduce inflammation and prevent complications such as restenosis of the dilated Eustachian tube. Patients were also instructed to perform the Valsalva maneuver twice daily, starting from the day of discharge. This involves exhaling against a closed airway to pop open the Eustachian tubes and helps maintain tube patency after surgery.

### 2.6. Follow-Up

Follow-up visits at the outpatient clinic were scheduled at 1, 3, and 6 months postoperatively. At these visits, patients underwent examinations such as otoscopy, tympanometry, and ETDQ-7 scoring to assess their healing process and outcomes. The 1-month follow-up visit is a crucial timepoint for identifying potential early complications such as persistent Eustachian tube obstruction or middle ear effusion requiring intervention. The later 3- and 6-month visits evaluated the sustained symptom relief and Eustachian tube function over time. Patients were also reminded to continue performing regular Valsalva maneuvers at home and report any concerning symptoms. Diligent postoperative monitoring and patient compliance with follow-up care are vital for ensuring optimal outcomes from BET procedures.

A range of pre- and postoperative data at 6 months were collected for each patient, including demographics such as age and sex, tympanic membrane condition, Eustachian tube function test scores, ETDQ-7 symptom scores, Eustachian tube mucosal inflammation scale, and the results of the Valsalva maneuver. All patients were followed for a minimum of 6 months postoperatively.

### 2.7. Statistical Analysis

The collected dataset was compiled and analyzed using SPSS version 22 statistical software (IBM Corp, Armonk, NY, USA). Continuous data were expressed as mean values with standard deviation (mean ± SD). Categorical data were expressed as percentages. The Kolmogorov–Smirnov test was performed to ascertain the probability distribution of the data. Since the resulting *p*-value exceeded 0.05, the data were found to conform to a normal distribution. Consequently, parametric tests were deemed suitable for evaluating statistical significance. To compare categorical variables between the BET alone (BET only) and BET with myringotomy (BET + M) groups, chi-square tests were utilized. For nonparametric continuous data, Mann–Whitney U tests were performed to assess the differences between groups. A *p*-value < 0.05 was considered statistically significant.

## 3. Results

A total of 95 patients underwent the procedure, of whom 51 underwent BET + M, and another 44 underwent BET only. The characteristics of sex and age are described in Table 1.

### 3.1. Tympanogram

Regarding the tympanometry results, abnormal tympanogram curves were more frequently observed preoperatively in both groups. In the BET + M group, 20 patients (39.2%) exhibited abnormal tympanograms preoperatively, all showing type C curves. Type C curves are considered abnormal. Postoperatively, eight patients still had abnormal type C tympanograms. Similarly, in the BET only group, 17 patients (38.6%) had abnormal preoperative tympanograms, again all type C. After surgery, nine patients continued showing type C abnormalities. The proportion of patients with persistent abnormal tympanometry after treatment was not significantly different between these two groups (*p* > 0.05) (Table 2).

### 3.2. Modified Inflation–Deflation Test

Regarding the modified inflation–deflation test, all patients in both groups exhibited abnormal results preoperatively, indicating impaired Eustachian tube function. At 6 months postoperatively, the BET without myringotomy group had eight patients (15.7%) with abnormal test findings, while the BET + M group had nine patients (20.5%) with continued dysfunction. The difference in the proportion of patients with persistent abnormal test results between the two groups was not statistically significant (*p* > 0.05) (Table 2).

### 3.3. ETDQ-7 Scores

In this study, we compared the ETDQ-7 symptom scores between the BET + M and BET only groups. The ETDQ-7 is a validated seven-item questionnaire assessing patient-reported Eustachian tube dysfunction severity. The mean preoperative 6-month ETDQ-7 score was similar between groups, measuring 27.9 ± 7.82 in the BET + M group and 28.3 ± 8.91 in the BET only group (*p* = 0.8162) (Table 1). At the 6-month postoperative follow-up, the BET + M group had a mean ETDQ-7 score of 18.7 ± 3.75, while the BET only group scored 17.9 ± 4.71 (*p* = 0.3594). The difference between groups was not statistically significant at either timepoint according to two-sample *t*-tests (*p* > 0.05) (Table 2).

### 3.4. Positive Valsalva Maneuver

At 6 months postoperatively, the Valsalva maneuver results were compared between the BET + M and BET only groups. Among the patients, 35 patients (58.8%) in the BET + M group and 28 patients (63.6%) in the BET only group had positive Valsalva maneuver results. There was no statistically significant difference between the two groups (*p* = 0.6665) (Table 2).

### 3.5. Eustachian Tube Mucosa Inflammation Scale

The Eustachian tube mucosa inflammation scale score was assessed and compared between the BET + M and BET only groups. The mean 6-month postoperative score for BET + M was 1.85 ± 0.63, and that for BET only was 1.82 ± 0.63 (*p* = 0.8564) (Table 1). After 6 months postoperatively, the mean score for BET + M was 1.78 ± 0.74, and that for BET only was 1.82 ± 0.67 (*p* = 0.7844) (Table 2). No statistically significant difference in the Eustachian tube mucosa inflammation scale score was found between the two groups.

### 3.6. Improvement of Eustachian Tube Function (Table 3)

Both the BET + M and the BET only groups demonstrated notable improvements in tympanometry results following surgery. In the BET + M group, the proportion of patients with normal type A tympanograms increased significantly from 31 (60.8%) preoperatively to 43 (84.3%) at 6 months postoperatively (*p* < 0.05). Similarly, in the BET only group, the percentage of patients with normal type A tympanograms rose substantially from 27 (61.4%) preoperatively to 35 (79.5%) at 6 months (*p* < 0.05). When comparing the proportion of patients showing tympanometric improvement between groups, no statistically significant differences were found at the 1-month (BET + M: 7 (13.7%) vs. BET only: 8 (18.2%), *p* = 0.5525), 3-month (BET + M: 11 (21.5%) vs. BET only: 8 (18.2%), *p* = 0.6807), or 6-month (BET + M: 12 (23.5%) vs. BET only: 8 (18.2%), *p* = 0.5238) follow-up timepoints.

**Table 3 jpm-13-01527-t003:** Comparison of improvement between groups during follow-up.

Variables	Improvement (1 Month)	Improvement (3 Months)	Improvement (6 Months)
Tympanogram			
BET + M	7 (13.7%)	11 (21.5%)	12 (23.5%)
BET only	8 (21.8%)	8 (21.8%)	8 (21.8%)
*p*-Value	0.5525	0.6807	0.5238
Modified inflation–deflation test results			
BET + M	37 (72.5%)	47 (92.1%)	49 (96.7%)
BET only	36 (81.8%)	41 (93.1%)	41 (93.1%)
*p*-Value	0.3354	1.0000	0.6600
ETDQ-7 score			
BET + M	47 (92.1%)	48 (94.1%)	45 (88.2%)
BET only	38 (86.3%)	38 (86.3%)	35 (79.5%)
*p*-Value	0.5056	0.2946	0.6431

The chi-square test was performed for categorical variables. M: myringotomy; BET: balloon Eustachian tuboplasty; ETDQ-7: Eustachian Tube Dysfunction Questionnaire-7.

The modified inflation–deflation test results improved markedly in both the BET + M and the BET only groups following surgery. In the BET + M group, the percentage of patients with normal test findings increased substantially from 0% preoperatively to 43 (84.3%) at 6 months postoperatively (*p* < 0.001). Similarly, in the BET only group, the proportion of patients with normal test results rose markedly from 0% preoperatively to 35 (79.5%) at 6 months (*p* < 0.001). When comparing the percentage of patients demonstrating improvement between groups, there were no statistically significant differences observed at the 1-month (BET + M: 37 (72.5%) vs. BET only: 36 (81.8%), *p* = 0.3354), 3-month (BET + M: 47 (92.1%) vs. BET only: 41 (93.1%), *p* = 1.0000), or 6-month (BET + M: 49 (96.7%) vs. BET only: 41 (93.1%), *p* = 0.6600) follow-up visits.

The mean ETDQ-7 score significantly improved in both groups after treatment. Specifically, in the BET + M group, the mean ETDQ-7 score improved from 27.9 ± 7.82 at baseline to 18.7 ± 3.75 at 6 months (*p* < 0.001). In the BET only group, the mean ETDQ-7 score improved from 28.3 ± 8.91 at baseline to 17.9 ± 4.71 at 6 months (*p* < 0.001). The difference in the amount of improvement between groups was not statistically significant (*p* = 0.3594). In terms of the proportion of patients showing improvement, a higher percentage of patients in the BET + M group improved compared to the BET only group at each postoperative timepoint, although the differences were not statistically significant (1 month: 92.1% vs. 86.3%, *p* = 0.5056; 3 months: 94.1% vs. 86.3%, *p* = 0.2946; 6 months: 88.2% vs. 79.5%, *p* = 0.6431).

## 4. Discussion

This study compared BET only versus BET + M for ETD treatment. Both groups showed significant ETDQ-7 score improvements without significant differences in scores, audiometry, endoscopy, or complications. These findings suggest that myringotomy does not provide added benefit to BET for ETD.

A comprehensive literature review was undertaken to summarize the current knowledge regarding ETD assessment and treatment options [19,20]. Various methods have been utilized to evaluate Eustachian tube function and diagnose ETD, including tympanometry, sonotubometry, and direct visualization or testing of the Eustachian tube [14,16,17]. These assessments provide objective evidence of ETD and can monitor treatment outcomes [1,15]. In terms of treatments, multiple medical and surgical options have been proposed for ETD through the years [14,21]. Conservative measures such as medications, autoinflation, and Valsalva maneuvers may be trialed first [19,22,23]. Other options range from minimally invasive balloon dilation procedures to more invasive surgeries such as myringotomy or tube insertion [7,8,17]. Each treatment carries its own risks and benefits [11,19,23]. Recent studies have explored emerging techniques such as BET as potentially effective ETD treatments with lower complication rates compared to traditional surgeries [6,19,23]. However, additional high-quality evidence is still needed to determine the optimal ETD treatments for different patient populations [20].

As a relatively new technique, BET has become increasingly popular as a minimally invasive surgical option for treating ETD [3,14]. Multiple studies have systematically reviewed the evidence on BET and found that it has similar efficacy to more invasive or complicated surgeries while carrying a lower risk of complications [6,16,23]. Specifically, BET demonstrates comparable improvement in subjective ETD symptoms and objective assessments of Eustachian tube function [19,24]. It also avoids risks such as persistent perforation, infection, bleeding, or scar tissue formation associated with techniques such as tube insertion or myringotomy [11,25]. Some potential advantages of BET include its quick recovery time and repeatability if needed [23]. While BET is a promising ETD treatment, varied patient outcomes highlight the need for further studies to determine which patients are most likely to benefit from this approach. Overall, BET demonstrates a favorable balance of efficacy and safety compared to traditional ETD surgeries.

Myringotomy is one of the oldest surgical techniques used to treat ETD [8,9]. It provides immediate relief by releasing the pressure and fluid buildup through an incision in the eardrum [2,26]. However, the benefits of myringotomy may only be temporary before the fluid reaccumulates [11,27,28]. Myringotomy also carries known risks of complications, including hearing loss, vertigo, infection, and persistent perforation of the eardrum [25]. The transient effects and risk profile have led to the decreased use of myringotomy in recent years compared to other options [16,19]. Techniques such as BET may more directly improve Eustachian tube dysfunction for longer-term results without damaging the tympanic membrane [14]. While myringotomy is still performed, current research suggests that it has questionable long-term efficacy and higher risks compared to emerging ETD treatments that focus on the physiological root cause [16].

An important consideration in ETD treatment is the complex relationship between objective assessments of Eustachian tube function and patient-reported symptom severity [1]. These two measures do not always correlate in a predictable linear fashion [29]. Some patients with essentially normal Eustachian tube function on testing may still experience significant ETD symptoms, while others with clearly abnormal test results may be relatively asymptomatic [23]. This discrepancy highlights the importance of using both objective and subjective measurements when evaluating ETD treatments [15,16]. Patient-reported symptom questionnaires provide crucial data on the real-world efficacy of ETD therapies in relieving symptoms that negatively impact quality of life [21]. Relying solely on objective function tests could overlook patients with persistent symptoms despite normal test results [17]. Our study used both ETDQ-7 symptom scores and Eustachian tube function testing to fully assess the outcomes.

In this study, we evaluated multiple outcome measures, both subjective and objective, to determine the efficacy of BET with or without myringotomy. The primary subjective outcome was a change in the ETDQ-7 questionnaire, a validated seven-item tool for assessing ETD symptoms. Our results showed that the addition of myringotomy did not improve the ETDQ-7 scores over BET only. We also assessed objective outcomes such as tympanometry, Eustachian tube function tests, otoscopy, and Valsalva maneuver results. Again, the addition of myringotomy did not improve these objective measures. Examining both subjective and objective factors provides a comprehensive view of how these procedures impact the patient experience of ETD symptoms, as well as physiologic Eustachian tube function [16]. Our findings suggest that the risks of myringotomy may not be justified, since it did not improve the subjective or objective ETD outcomes beyond BET only.

This study provides further evidence that BET has an acceptable safety profile with relatively low risk when proper precautions are taken. However, any surgery on delicate ear and nasal structures still carries inherent risks requiring careful mitigation [14]. Potential complications of BET parallel those seen with related balloon dilation procedures, including barotrauma, mucosal injury, bleeding, infection, and damage to the surrounding anatomy [3,30]. Proper patient selection is critical, as anatomical abnormalities such as septal deviations or hypotrophic tubes increase complication risks and should be ruled out preoperatively [22]. Careful and gradual inflation of the balloon under endoscopic visualization can also reduce the risks during BET [18].

In this study, Eustachian tube mucosal lacerations affected 13.7% of the BET + M group versus 9.1% of the BET only group. Prior research found mucosal tears in 17–57% of patients, often resolving without intervention [27]. Concurrent myringotomy did not increase the mucosal trauma versus BET only. One case of emphysema occurred in the BET only group, consistent with the 0–3% incidence range reported previously [25]. Technique adjustments such as avoiding overinflation can further minimize barotrauma events [21]. No instances of balloon placement failure or uncontrolled bleeding occurred. However, one BET + M patient experienced persistent tympanic membrane perforation requiring later tympanoplasty. Prior studies found persistent perforations in 0–13% of patients after myringotomy [29]. Postoperative monitoring remains vital to detect rare issues such as osteitis or Eustachian tube obstruction [3]. Overall, the safety findings align with the existing literature on the complication profile of BET and myringotomy. Although not completely risk-free, data support BET as less invasive than historical ETD surgery options when performed carefully in appropriate patients.

Ventilation tube insertion represents a traditional surgical treatment for chronic ETD and otitis media with effusion (OME), providing an alternative means of ventilating the middle ear space [31]. Multiple studies have shown that tube insertion can improve hearing, reduce effusion, and relieve any associated symptoms in both pediatric and adult ETD patients [32,33,34]. However, ventilation tubes represent an indirect treatment option that does not address the underlying pathology of ET. Tubes may need to be replaced multiple times, and patients can experience complications such as otorrhea, tympanosclerosis, and persistent perforation [35]. Recent research suggests emerging techniques such as BET may have comparable effectiveness to ventilation tubes for certain ETD patients while more directly improving ET structure and function [32,36,37]. Still, tube insertion remains a validated option for managing ETD and OME. While invasive, it provides clinically meaningful benefits for many patients and should be considered alongside newer approaches.

In summary, this retrospective study did not find significant differences in the subjective or objective ETD outcomes between BET + M and BET only. The addition of myringotomy did not improve the results based on the ETDQ-7 scores, tympanometry, modified inflation–deflation test, or other measures. Furthermore, myringotomy did not reduce procedural risks such as emphysema, mucosal laceration, nasal bleeding, or BET failure compared to BET only. However, myringotomy did add the risk of persistent tympanic membrane perforation, requiring later tympanoplasty in one BET + M patient. Further large-scale randomized controlled trials with long-term follow-up are needed to definitively compare these techniques and patient populations. One limitation of this study was that it did not include a group receiving myringotomy alone for comparison. Comparing outcomes across BET alone, BET with myringotomy, and myringotomy alone would have provided more definitive evidence regarding the additive effect of myringotomy. Additional limitations include the single-center design and lack of subgroup analysis based on ETD severity. However, within the constraints of this retrospective study, our findings suggest that BET only sufficiently and safely improves ETD symptoms without the need for myringotomy in this cohort. More research is needed to determine whether certain ETD subtypes may benefit from combining approaches.

## 5. Conclusions

In conclusion, this study found that both BET only and BET + M improved ETD symptoms without significant differences. BET only did not increase complications compared to BET + M. However, BET + M incurred risks such as persistent tympanic membrane perforation without additional therapeutic advantages. These preliminary findings support BET only as an efficacious and safe alternative to conventional ETD surgeries. Further large randomized controlled trials should identify optimal candidates for BET only versus combined approaches. With evidence, BET could transform ETD treatment algorithms.

## Figures and Tables

**Table 1 jpm-13-01527-t001:** Demographic and baseline characteristics of patients.

Variables	BET + M (*n* = 51)	BET Only (*n* = 44)	*p*-Value
Sex (male; female)	28; 23	27; 17	0.5402
Age	49.7 ± 13.85	50.78 ± 14.5	0.7116
Tympanogram A	31 (60.8%)	27 (61.4%)	
Tympanogram B	0 (0%)	0 (0%)	
Tympanogram C	20 (39.2%)	17 (38.6%)	0.9540
Modified inflation–deflation test results			
Good	0 (0%)	0 (0%)	
Fair	18 (35.3%)	19 (43.1%)	
Poor	33 (64.7%)	25 (56.9%)	0.5277
ETDQ-7 score	27.9 ± 7.82	28.3 ± 8.91	0.8162
Patients with positive Valsalva maneuver results	14 (27.4%)	11 (25%)	0.8193
Eustachian tube mucosa inflammation scale score	2.97 ± 0.77	2.86 ± 0.92	0.5274

The chi-square test was performed for categorical variables. The Mann–Whitney U test was performed for continuous variables. M: myringotomy; BET: balloon Eustachian tuboplasty; ETDQ-7: Eustachian Tube Dysfunction Questionnaire-7.

**Table 2 jpm-13-01527-t002:** Results at 6 months postoperatively.

Variables	BET + M (*n* = 51)	BET Only (*n* = 44)	*p*-Value
Tympanogram A	43 (84.3%)	35 (79.5%)	
Tympanogram B	0 (0%)	0 (0%)	
Tympanogram C	8 (15.7%)	9 (20.5%)	0.5455
Modified inflation–deflation test results			
Good	43 (84.3%)	35 (79.5%)	
Fair	0 (0%)	0 (0%)	
Poor	8 (15.7%)	9 (20.5%)	0.5988
ETDQ-7 score	18.7 ± 3.75	17.9 ± 4.71	0.3594
Patients with positive Valsalva maneuver results	35 (58.8%)	28 (63.6%)	0.6665
Eustachian tube mucosa inflammation scale score	1.78 ± 0.74	1.82 ± 0.67	0.7844

The chi-square test was performed for categorical variables. The Mann–Whitney U test was performed for continuous variables. M: myringotomy BET: balloon Eustachian tuboplasty; ETDQ-7: Eustachian Tube Dysfunction Questionnaire-7.

## Data Availability

The datasets generated during and/or analyzed during the current study are available from the corresponding author on reasonable request.

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
