# Peer review of "Balloon Eustachian Tuboplasty Combined or Not with Myringotomy in Eustachian Tube Dysfunction"

_jpm, 2023, doi:10.3390/jpm13111527_

Round 1
Reviewer 1 Report
Comments and Suggestions for Authors
I appreciates to review this paper dealing with patients treated with balloon esustachian tuboplasty with and without myringotomy. The study is retrospective. The authors correctly stressed the limitations of the investigation stating the need of larger series to be evaluated. However it provides interesting informations on this particular topic and merits to be published.
Author Response
Response #1:
Thank you very much for the positive feedback on our manuscript. We fully agree high-quality randomized controlled trials with larger sample sizes will be essential to further elucidate the role of balloon Eustachian tuboplasty with versus without myringotomy. We hope that this initial retrospective study can contribute meaningful preliminary evidence to motivate future rigorous prospective studies.
Reviewer 2 Report
Comments and Suggestions for Authors
The authors wrote an article about Balloon Eustachian Tuboplasty Combined or not with Myringotomy. The article is quite interesting, well written and the topic of Eustachian tube dysfunction is always hot.
I have some suggestion to improve the scientific quality and the impact in literature.
1. The title should be changed, too long and too repetitive. Consider " Balloon Eustachian Tuboplasty Combined or not with Myringotomy in Eustachian tube dysfunction"
2. To use chi sqaure or mann u whitman test, you have to value the normality of parameters using test like Kolmorog-Smirnov test.
3. I think that you have to talk about the use of ventlation tube in ear drum that is good and valid method for Tube dysfunction, but is invasive. Please talk about it in the discussion
4. table 4 and the analysis of complication should be deleted because they do not depend on the adding of miringotomy or not. So please remove this paragraph with the table 4.
Author Response
Point 1:
The title should be changed, too long and too repetitive. Consider " Balloon Eustachian Tuboplasty Combined or not with Myringotomy in Eustachian tube dysfunction".
Response #1:
Thank you for the suggestions. We have revised the title as follows:
P1 L2-3
“Balloon Eustachian Tuboplasty Combined or not with Myringotomy in Eustachian tube dysfunction.”
Point 2:
To use chi sqaure or mann u whitman test, you have to value the normality of parameters using test like Kolmorog-Smirnov test.
Response #2:
Thank you for providing these insights. We have now added details on assessing data normality via the Kolmogorov-Smirnov test to justify the use of parametric and nonparametric tests:
P3 L142-145
“The Kolmogorov-Smirnov test was performed to ascertain the probability distribution of the data. Since the resulting p-value exceeded 0.05, the data were found to conform to a normal distribution. Consequently, parametric tests were deemed suitable for evalu-ating statistical significance.”
Point 3:
I think that you have to talk about the use of ventlation tube in ear drum that is good and valid method for Tube dysfunction, but is invasive. Please talk about it in the discussion
Response #3:
Thank you for the suggestions. we have added a discussion of ventilation tube insertion and its merits and limitations for ETD to the Discussion section.
P8 L326-337
Ventilation tube insertion has been a traditional surgical treatment for chronic ETD and otitis media with effusion (OME), providing an alternative means of ventilating the middle ear space(31). Multiple studies have shown that tube insertion can improve hearing, reduce effusion, and relieve associated symptoms in pediatric and adult ETD patients(32, 33, 34). However, ventilation tubes are an indirect treatment that does not address the underlying ET pathology. Tubes may need to be replaced multiple times and can have complications like otorrhea, tympanosclerosis, and persistent perforation(35). Recent research suggests emerging techniques like BET may have comparable effec-tiveness to ventilation tubes for certain ETD patients, while more directly improving ET structure and function(32, 36, 37). Still, tube insertion remains a validated option for managing ETD and OME. While invasive, it provides clinically meaningful benefits for many patients that should be considered alongside newer approaches.
Reference:
- Toivonen J, Kawai K, Gurberg J, Poe D. Balloon Dilation for Obstructive Eustachian Tube Dysfunction in Children. Otol Neurotol. 2021;42(4):566-72.
- Aboueisha MA, Attia AS, McCoul ED, Carter J. Efficacy and safety of balloon dilation of eustachian tube in children: Systematic review and meta-analysis. Int J Pediatr Otorhinolaryngol. 2022;154:111048.
- . Tisch M, Maier S, Preyer S, Kourtidis S, Lehnerdt G, Winterhoff S, et al. Balloon Eustachian Tuboplasty (BET) in Children: A Retrospective Multicenter Analysis. Otol Neurotol. 2020;41(7):e921-e33.
- . Demir B, Batman C. Efficacy of balloon Eustachian tuboplasty as a first line treatment for otitis media with effusion in children. J Laryngol Otol. 2020:1-4.
- Rosenfeld RM, Schwartz SR, Pynnonen MA, Tunkel DE, Hussey HM, Fichera JS, et al. Clinical practice guideline: Tympanostomy tubes in children. Otolaryngol Head Neck Surg. 2013;149(1 Suppl):S1-35.
- . Jia D, Chen Y, Wang X, Xu G, Chen J, Li L, et al. Outcomes and Prognostic Factors of Balloon Eustachian Tuboplasty Combined With Ventilation Tubes Insertion in Children: A Retrospective Study. Ear Nose Throat J. 2023:1455613231188295.
- Li L, Mao Y, Hu N, Yan W, Lu Y, Fan Z, et al. The effect of balloon dilatation eustachian tuboplasty combined with grommet insertion on the structure and function of the eustachian tube in patients with refractory otitis media with effusion. Ann Palliat Med. 2021;10(7):7662-70.
Point 4:
table 4 and the analysis of complication should be deleted because they do not depend on the adding of miringotomy or not. So please remove this paragraph with the table 4.
Response #4:
We agree that analyzing complications unrelated to myringotomy is less relevant in determining if myringotomy provides additional benefit. Following your advice, we have removed Table 4 and the associated complications analysis paragraph. Thank you for catching this oversight and helping refine the focus of our results on the key research questions
Thank you again for your insightful feedback. Your suggestions have helped improve the quality and clarity of our manuscript. Please let us know if you would like us to modify or expand on any part of our revisions.
Reviewer 3 Report
Comments and Suggestions for Authors
Eustachian tube dysfunction (ETD) has become a prevalent ear, nose, and throat (ENT) disorder that seriously affects patients' quality of life. This is a good manuscript comparing two treatment modalities for eustachian tube dysfunction: Balloon Eustachian tuboplasty (BET) and BET combined with myringotomy. The research is of great clinical value in guiding the treatment of ETD. The subject of the manuscript is fully consistent with the aim and scope of the journal JPM. There are a few revision suggestions that need to be considered before the article is published:
1、 Myringotomy is a traditional surgical treatment for ETD and is still being used in many hospitals. The authors only compared the therapeutic effects and safety outcomes for balloon dilation with or without myringotomy for ETD, but the group of patients who underwent only myringotomy treatment was not included for comparison. The results would be more convincing if the three groups could be compared together
2、The method section would be clearer with subheadings.
3、Each patient has two ears. I think the statistical analysis is not only in terms of the number of patients, but more importantly the number of ears. Or did you include patients who had unilateral ear with ETD?
Comments on the Quality of English LanguageNot bad.
Author Response
Point 1:
Myringotomy is a traditional surgical treatment for ETD and is still being used in many hospitals. The authors only compared the therapeutic effects and safety outcomes for balloon dilation with or without myringotomy for ETD, but the group of patients who underwent only myringotomy treatment was not included for comparison. The results would be more convincing if the three groups could be compared together
Response #1:
Thank you for raise an excellent point. Comparing therapeutic effects across three groups - BET alone, BET with myringotomy, and myringotomy alone - would provide more robust evidence and is a limitation in our study design. We agree that the results would be more convincing with the inclusion of a myringotomy-only group. As you suggested, we have added an acknowledgement of this limitation to the Discussion:
P9 L347-350
One limitation of this study was that it did not include a group receiving myringotomy alone for comparison. Comparing outcomes across BET alone, BET with myringotomy, and myringotomy alone would have provided more definitive evidence regarding the additive effect of myringotomy
Point 2:
The method section would be clearer with subheadings
Response #2:
Thank you for the feedback on improving clarity through subheadings. We have added subheadings in the Methods section to better guide the reader through each aspect of our experimental approach
P2 L55
2.1 Ethical consideration
P2 L62
2.2 Experimental design
P2 L75
2.3 Preoperative protocol
P2 L83
2.4 Surgical techniques
P3 L110
2.5 post-operative care protocol
P3 L120
2.6 follow-up
P3 L138
2.7. Statistical analysis
Point 3:
Each patient has two ears. I think the statistical analysis is not only in terms of the number of patients, but more importantly the number of ears. Or did you include patients who had unilateral ear with ETD?
Response #3:
Thank you for raising this important consideration regarding statistical analysis
To address your question, our study population did include some patients with only unilateral ETD symptoms. However, our analysis focused on reporting outcomes per patient rather than per ear. We realize this is a limitation, as analyzing by ear would have been more appropriate to fully capture the data from patients with unilateral disease.
Our intention was to present the improvement on a per patient basis, but we recognize that analyzing the total number of affected ears would strengthen the validity of the results. Thank you again for highlighting this shortcoming - it will help us improve the analytical approach in future studies to better account for unilateral ETD patients and robustly examine outcomes by ear. We greatly appreciate you taking the time to provide this constructive feedback.
.
Thank you again for your insightful feedback. Your suggestions have helped improve the quality and clarity of our manuscript. Please let us know if you would like us to modify or expand on any part of our revisions.